# Acceptance of COVID-19 Vaccination among Healthcare and Non-Healthcare Workers of Hospitals and Outpatient Clinics in the Northern Region of Slovakia

**DOI:** 10.3390/ijerph182312695

**Published:** 2021-12-02

**Authors:** Romana Ulbrichtova, Viera Svihrova, Maria Tatarkova, Henrieta Hudeckova, Jan Svihra

**Affiliations:** 1Department of Public Health, Jessenius Faculty of Medicine in Martin, Comenius University in Bratislava, Mala Hora 11149/4B, 036 01 Martin, Slovakia; romana.ulbrichtova@uniba.sk (R.U.); maria.tatarkova@uniba.sk (M.T.); henrieta.hudeckova@uniba.sk (H.H.); 2Clinic of Urology, Jessenius Faculty of Medicine in Martin, Comenius University in Bratislava, Kollarova 2, 036 59 Martin, Slovakia; jan.svihra@uniba.sk

**Keywords:** healthcare workers, COVID-19, vaccination, hesitancy

## Abstract

The purpose of this study was to analyse attitudes, motivation, and reasons for hesitancy toward COVID-19 vaccination among healthcare workers (HCWs) in northern Slovakia. A cross-sectional study was conducted between 30 August 2021 and 30 September 2021. An anonymous questionnaire was administered. The study was completed by 1277 employees. Multivariate logistic regression was used to identify predictors of COVID-19 vaccination status. A total of 1076 (84.3%) were vaccinated, and 201 (15.7%) were unvaccinated. Physician job type (OR = 1.77; CI95 1.13–2.78), history of COVID-19 (OR = 0.37; CI95 0.26–0.37), influenza vaccination at any time (OR = 1.97; CI95 1.12–3.46), compulsory vaccination for HCWs (OR = 9.15; CI95 2.92–28.62), and compulsory vaccination for selected groups (OR = 9.71; CI95 2.75–34.31) were the predictors significantly associated with COVID-19 vaccination acceptance. Non-physician HCWs, employees in hospitals, and employees without a history of COVID-19 significantly more distrusted the efficacy of vaccines against COVID-19. Results of our study confirmed that physicians have higher vaccination rates and lower hesitance to get vaccinated than non-physician HCWs. HCWs play an important role in influencing vaccination decisions and can be helpful in vaccine advocacy to the general public.

## 1. Introduction

Since 11 March 2020, when the World Health Organisation (WHO) declared the epidemic of coronavirus disease 2019 (COVID-19) as a pandemic, healthcare priorities have changed [1]. Healthcare workers (HCWs) are considered the most important members for any healthcare system in the fight against COVID-19. We recorded a high morbidity and mortality rates among HCWs [2].

Ending the COVID-19 pandemic will require unprecedented collective action. Non-pharmaceutical interventions (handwashing, face masks, social distancing, travel restrictions, etc.) in combination with safe and effective vaccines and the implementation of global and national vaccination programmes are the only available tools to allow humanity to return to pre-pandemic life and to protect people against COVID-19 [3].

In the Slovak Republic (SR), from 3 January 2020 to 16 November 2021, there have been 1,020,022 confirmed cases of COVID-19 and 13,644 deaths reported to the WHO. In the SR, vaccinations started on 26 December 2020. The Ministry of Health of the SR has presented 12 phases within the national vaccination programme, which clearly determine the order of vaccination. Vaccination among HCWs was in the first phase, and the country prioritised the vaccination of HCWs who were in contact with COVID-19 patients [4,5].

Currently, several COVID-19 vaccines are publicly available in the SR. Achieving collective immunity is a necessary step to end the pandemic. Mathematical models show that if the COVID-19 vaccine is 80% effective, then the coverage must be at least 75% to eliminate an ongoing pandemic [6]. Globally, there have been 3,223,744,666 persons fully vaccinated, and in the SR there have been 2,426,386 persons [4].

Therefore, it is extremely important to keep track of the public’s views on vaccination, especially the views on, and acceptance of, the vaccine among HCWs. Some countries have gradually introduced compulsory vaccination for selected population groups. The United Kingdom government recently introduced new legislation, effective 11 November 2021, that requires people working in care homes to be COVID-19 vaccinated [7]. Italy became the first country in Europe to make vaccination against COVID-19 mandatory for HCWs [8]. Subsequently, France and Greece followed Italy’s precedent [9].

The aim of this study was to analyse the attitudes, motivation, and reason for hesitancy towards COVID-19 vaccination among HCWs and non-HCWs in hospitals and outpatient healthcare clinics in northern Slovakia.

## 2. Materials and Methods

### 2.1. Study Population

A cross-sectional study was conducted between 30 August and 30 September 2021 by administering an anonymous questionnaire distributed to individual e-mail addresses. We addressed all HCWs and non-HCWs belonging to hospitals and general and specialised outpatient healthcare clinics in northern Slovakia (*n* = 4268). Data were collected from physicians (*n* = 1455), non-physician HCWs (*n* = 2166), and non-HCWs (*n* = 647) [10]. Employees were instructed not to complete the questionnaire more than once. The study was completed by 1277 of 4268 employees from hospitals and general and specialised outpatient healthcare clinics. The employees who had received at least one dose of the COVID-19 vaccine were included in the vaccinated group (*n* = 1076). The employees who had not received any vaccine were included in the unvaccinated group (*n* = 201) (Figure 1).

### 2.2. Survey Questionnaire

A Microsoft Forms questionnaire was used, which was adopted from Stepanek et al. with their consent [11]. Socio-demographic characteristics were collected among all respondents. Motives for vaccination were collected only among vaccinated respondents. Reasons for hesitancy were collected only among unvaccinated respondents (Appendix A).

### 2.3. Statistical Analyses

All statistical analyses were carried out using the software SPSS 24, R version 4.0.2, and EPI Info 7 using methods of descriptive statistics, including a chi-square test for pairwise comparisons [12]. Student’s *t*-test was used for comparison between two means for quantitative variables. Comparisons of fear levels were performed using the analysis of variance (ANOVA). Multivariate logistic analysis was performed for prediction of COVID-19 vaccination. Odds ratios with 95% confidence intervals were used to identify statistically significant differences between the vaccinated and unvaccinated groups. A *p* value of less than 0.05 was considered statistically significant. Binomial logistic regression was performed to examine the relationships between vaccine acceptance (being “vaccinated” or “unvaccinated” as a response variable) and other characteristics included in the responses of all employees (explanatory variables).

## 3. Results

A total of 4268 respondents were given questionnaires, and 1277 respondents were considered for the analysis, of which 1076 (84.3%) were vaccinated and 201 (15.7%) unvaccinated. Among vaccinated participants, 93.1% were physicians, 76.9% non-physician HCWs, and 76.5% non-HCWs. Among all vaccinated participants, 96.8% had received the Comirnaty vaccine (Pfizer-BioNTech), 1.3% had received the Spivax (Moderna) vaccine, 1% had received the Vaxzevria (AstraZeneca) vaccine, 0.5% had received the Janssen (Johnson & Johnson) vaccine, and 0.4% had received the Sputnik vaccine. The median age was 48.3 years (range 19–87 years). The average years of work were 21.1 years (range 0–62), and 78.1% were female. Among all respondents in the study, 582 (45.6%) were physicians, 542 (42.4%) were non-physician HCWs, and 153 (11.9%) were non-HCWs. The socio-demographic characteristics of the participants are shown in Table 1.

The average age and the average job duration of vaccinated employees were higher compared to the unvaccinated. Unvaccinated participants showed a significantly lower level of fear (3.7 ± 2.4) than vaccinated participants (6.1 ± 2.7). More than 93% of total physicians were vaccinated. More than 27% of respondents (*n* = 354) reported to have had a previous infection from SARS-CoV-2 (Table 1).

When we performed multivariate logistic regression analysis, physician job type, compulsory vaccination for HCWs, and compulsory vaccination for selected groups were the predictors significantly associated with COVID-19 vaccination acceptance (Figure 2).

Motives to be vaccinated against COVID-19 are shown in Table 2. The effort to prevent the spread of COVID-19 during the performance of profession was a significantly stronger motivation for physicians than for non-physician HCWs. 

Reasons for hesitancy to get vaccinated against COVID-19 are shown in Table 3. The top reasons for refusing vaccination were concerns about the side effects and concerns about efficacy of the COVID-19 vaccine.

Table 4 display the attitudes towards compulsory vaccination for three groups: only HCWs, selected population groups, and the whole population. Physicians were more inclined to introduce compulsory vaccination for all three groups than non-physician HCWs (*p* < 0.001). Almost 65% of vaccinated physicians were agreeable towards introduction of compulsory vaccination among HCWs. Physicians play a significant role in influencing vaccination decisions.

## 4. Discussion

The B.1.617.2 (delta) variant was first detected in India in December 2020. It became the most commonly reported variant in the country starting in mid-April 2021 and has now been detected across the globe. Currently, the delta variant is responsible for a sharp increase in cases of COVID-19 disease [13]. To bring this pandemic to an end, a large share of the world needs to be immune to the virus. The safest way to achieve this is through vaccination. For this reason, vaccination together with other non-pharmacological interventions are necessary to protect people against COVID-19 and end the COVID-19 pandemic. Vaccines have been found to be highly efficacious at preventing symptomatic disease, as shown by clinical trials and real-world evidence [14,15].

The total number of people who have received at least one dose of the vaccine varies around the world. In the middle of November 2021, the countries with highest vaccination rates in the world were United Arab Emirates (88.20%), Portugal (87.56%), Singapore (86.25%), and Spain (80.08%). Slovakia is one of the European countries with a lower vaccination rate, as only 42.60% are fully vaccinated [4]. This low vaccination rate is the result of a weak vaccination strategy. Solidarity based vaccination strategies can only be effective if people have enough reliable information and consider all risks related to vaccination as low. Currently, Slovakia has not introduced any type of compulsory vaccination. Fortunately, with recent tightening of measures for the unvaccinated, we are seeing a slight increase in the vaccination rate [4,5].

The coronavirus pandemic has concurrently raised the issue of postponing or cancelling of elective surgeries. Ultimately, this has a significant impact on patients, causing distress to the population, and negative consequences for all healthcare systems. Furthermore, postponement of elective surgery should be a carefully considered step after weighing the dangers of transmitting the infection to the patient compared to morbidity and mortality incurred by postponing or cancelling surgery. The study of the COVIDSurg Collaborative estimated the total number of elective surgeries that would be cancelled worldwide during the 12 weeks. The best estimate was that 28,404,603 surgeries would be cancelled or postponed globally. In Slovakia, 4644 surgeries would have been cancelled per week during the COVID-19 pandemic [16,17,18].

Attitudes towards COVID-19 vaccines have become a challenge for HCWs. They play an important role in influencing vaccination decisions and can be helpful in vaccine advocacy to the general public [19]. General COVID-19 vaccine hesitancy among HCWs is common, and the prevalence among HCWs has ranged from 4.3% to 72% (average = 22.5%) [20]. At the time, vaccination coverage among our participants reached approximately 84%, of which 93.1% was among physicians, 76.9% among non-physician HCWs, and 76.5% among non-HCWs. Results of a similar published study showed high acceptance of vaccination among HCWs, but not all of them wanted to receive the vaccine. The acceptance rate of HCWs was significantly higher than of non-HCWs (76.9% vs. 56.2%) [21]. Interestingly, results of a German study showed 92% acceptance of COVID-19 vaccination among HCWs [22]. Our findings were consistent with recently published studies worldwide with acceptance rates between 31% and 86% [11,23,24,25].

In general, we know that women are more represented in health services than men. In Slovakia, at the end of 2019, there were 110,778 persons working in the health services, of which 23,913 (21,6%) were men and 86,865 (78,4%) were women [26]. Non-physician HCWs, especially nurses, were the most affected occupations infected by COVID-19. In our study, non-physician HCWs were less motivated to get vaccinated against COVID-19 than physicians. These results were also confirmed by authors of other studies [23,27]. The study in Cyprus observed around 70% of respondents (nurses and midwives) were undecided to accept COVID-19 vaccine [27].

In several studies, the leading reasons for refusing vaccination were concerns about the side effects and concerns about efficacy for COVID-19 vaccine [11,27,28]. More than 68% of our participants were concerned about the adverse effects of the COVID-19 vaccine, and 40.8% were concerned about its effectiveness.

Hospitals and outpatient clinics increase the risk of infection both for HCWs and patients, and also for their families. Telemedicine was not used in Slovakia before the pandemic. The patient’s examination took place either in a medical facility or at the patient’s home. Due to these facts and the fact that non-essential medical activities are often limited during a pandemic, the use of telemedicine has become an important way of managing patients’ healthcare. Telemedicine has been used abroad in the past, but during the pandemic there are more cases where specialists from various medical disciplines have started using telemedicine to care for patients. One of the several benefits of telemedicine is the reduction of the risk of COVID-19 infection through social distancing [29,30].

The priority and mission of physicians, as well as non-physician HCWs, are to promote vaccines and educate patients about the benefits of vaccination, emphasising that the benefits outweigh the risks and that each vaccine has been studied in detail to determine its safety profile.

### Limitations and Strengths

The first limitation of this study is due to the non-standardised questionnaire, which could lead to problems in interpreting results. We also explain the relatively low return rate of the questionnaire by the fact that employees who refuse vaccination did not fill in the questionnaire. This study is the first study to evaluate the COVID-19 vaccine hesitancy among HCWs in the SR. Therefore, the results can help to understand the reasons for refusing vaccination and focus on them by taking actions.

## 5. Conclusions

Physicians play a significant role in influencing vaccination decisions and can be helpful in vaccine advocacy to the general public. Our results confirmed that physicians have higher vaccination rates and lower hesitancy to get vaccinated than non-physician HCWs. They state the reasons for refusing vaccination against COVID-19 as concerns about efficacy and potential side effects. For this reason, it is necessary to focus on the correct interpretation of the vaccination strategy among HCWs, especially among non-physician HCWs.

## Figures and Tables

**Figure 1 ijerph-18-12695-f001:**
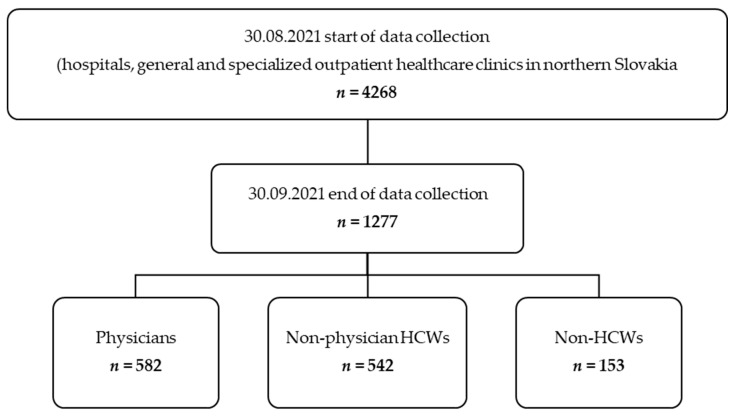
Process of data collection.

**Figure 2 ijerph-18-12695-f002:**
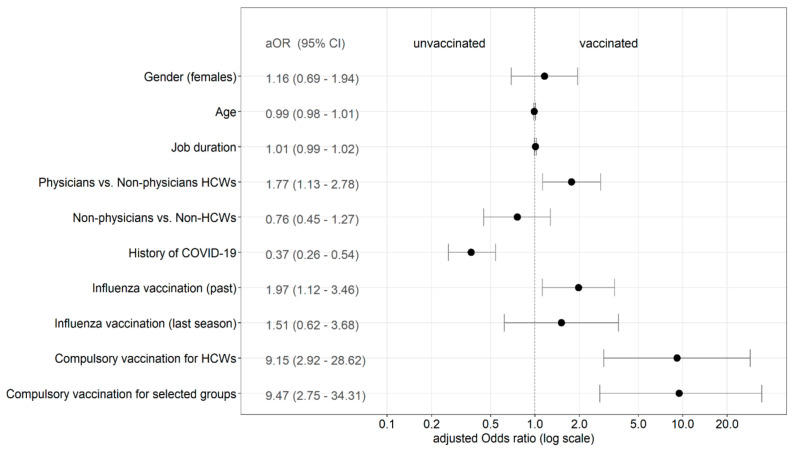
Predictors of COVID-19 vaccination (multivariate logistic regression analysis). Note: HCWs, healthcare workers; OR, Odds ratio; CI, confidence interval.

**Table 1 ijerph-18-12695-t001:** Socio-demographic characteristics of participants.

	Total (*n* = 1277)	Vaccinated (*n* = 1076)	Unvaccinated (*n* = 201)	*p* Value
Age (years; average ± SD)	48.3 ± 12.6	48.7 ± 12.6	46.3 ± 12.2	0.014 *
Job duration (years; average ± SD)	21.1 ± 15.1	21.9 ± 15.0	17.1 ± 14.8	<0.001 *
Level of fear of COVID-19 (average ± SD)	5.7 ± 2.7	6.1 ± 2.7	3.7 ± 2.4	<0.001 *
Male N (%) Female N (%)	280 997	249 (88.9) 827 (82.9)	31 (11.1) 170 (17.1)	0.016 ** 0.016 **
Physicians N (%) Non-physician HCWs N (%) Non-HCWs N (%)	582 542 153	542 (93.1) 417 (76.9) 117 (76.5)	40 (6.9) 125 (23.1) 36 (23.5)	<0.001 ** <0.001 ** 0.005 **
Hospitals N (%) Outpatient healthcare clinic N (%)	745 532	592 (79.5) 484 (91.0)	153 (20.5) 48 (9.0)	<0.001 ** <0.001 **
History of COVID-19 N (%)	354	250 (70.6)	104 (29.4)	<0.001 **
Influenza vaccinated at any time in the past N (%) Influenza vaccinated last season N (%)	477 283	449 (94.1) 274 (96.8)	28 (5.9) 9 (3.2)	<0.001 ** <0.001 **

Note: HCWs, healthcare workers; * *p* < 0.05 (Student’s *t*-test); ** *p* < 0.05 (chi-square test).

**Table 2 ijerph-18-12695-t002:** Motives to get vaccinated against COVID-19 (vaccinated group, N = 1076) according to selected variables (occupation and history of COVID-19).

Motives	Physicians N = 542	Physicians N = 542	Non-Physician HCWs N = 417	Hospitals N = 592	History of COVID-19 N =250
	Non-Physician HCWs N = 417	Non-HCWs N = 117	Non-HCWs N = 117	Outpatient HC N =484	No history of COVID-19 N = 826
Concerns about COVID-19 itself	380 (70.1%) *	380 (70.1%) *	233 (55.9%)	346 (58.5%) *	137 (54.8%) *
	233 (55.9%)	70 (59.8%)	70 (59.8%)	337 (69.6%)	546 (66.1%)
An effort to prevent the spread of COVID-19 during the performance of my profession	474 (87.5%)	474 (87.5%) *	347 (83.2%)	492 (83.1%)	206 (82.4%)
	347 (83.2%)	91 (77.8%)	91 (77.8%)	420 (86.8%)	706 (85.5%)
An effort to protect family members	393 (72.5%)	393 (72.5%)	287 (68.8%)	404 (68.2%) *	167 (66.8%)
	287 (68.8%)	81 (69.2%)	81 (69.2%)	357 (73.8%)	594 (71.9%)
Being exempted from restrictive anti-pandemic measures after vaccination	160 (29.5%)	160 (29.5%)	126 (30.2%)	174 (29.4%)	82 (32.8%)
	126 (30.2%)	29 (24.8%)	29 (24.8%)	141 (29.1%)	233 (28.2%)
Others	37 (6.8%)	37 (6.8%)	34 (8.1%)	40 (6.8%)	18 (7.2%)
	34 (8.1%)	7 (5.9%)	7 (5.9%)	38 (7.9%)	60 (7.3%)

Note: HCWs, healthcare workers; Outpatient HC, outpatient healthcare clinics; * *p* < 0.05 (Student’s *t*-test); multiple-choice options.

**Table 3 ijerph-18-12695-t003:** Reasons for personal COVID-19 vaccine hesitancy (unvaccinated group, N = 201) according to selected variables (occupation and history of COVID-19).

Reasons	Physicians N = 40	Physicians N = 40	Non-Physician HCWs N =125	Hospitals N = 153	History of COVID-19 N = 104
	Non-Physician HCWs N = 125	Non-HCWs N = 36	Non-HCWs N = 36	Outpatient HC N = 48	No History of COVID-19 N = 97
I am not afraid of COVID-19, its course and consequences	5 (12.5%)	5 (12.5%)	15 (12.0%)	15 (9.8%)	10 (9.6%)
	15 (12.0%)	3 (8.3%)	3 (8.3%)	5 (10.4%)	13 (13.4%)
I do not find getting infected with COVID-19 likely	0 (0.0%)	0 (0.0%)	9 (7.2%)	8 (5.2%)	2 (1.9%)
	9 (7.2%)	0 (0.0%)	0 (0.0%)	1 (2.1%)	7 (7.2%)
I do not trust the efficacy of vaccines against COVID-19	11 (27.5%) *	11 (27.5%)	58 (46.4%)	68 (44.4%) *	34 (32.7%) *
	58 (46.4%)	13 (36.1%)	13 (36.1%)	7 (15.6%)	48 (49.5%)
I have concerns about the safety and side effects of vaccines against COVID-19	28 (70.0%)	28 (70.0%)	86 (68.8%)	112 (73.2%) *	66 (63.5%)
	86 (68.8%)	28 (77.8%)	28 (77.8%)	25 (52.1%)	71 (73.2%)
I went through COVID-19 and assume lasting immunity against disease	12 (30.0%)	12 (30.0%)	43 (34.4%)	55 (35.9%)	68 (65.4%) *
	43 (34.4%)	14 (38.9%)	14 (38.9%)	14 (29.2%)	1 (1.0%)
I have contraindications of expect a complicated vaccination course in my case	10 (25.0%)	10 (25.0%)	30 (24.0%)	33 (21.6%)	22 (21.2%)
	30 (24.0%)	8 (22.2%)	8 (22.2%)	15 (31.3%)	26 (26.8%)
Others	15 (37.5%) *	15 (37.5%) *	19 (15.2%)	24 (15.7%) *	20 (19.2%)
	19 (15.2%)	6 (16.7%)	6 (16.7%)	16 (33.3%)	20 (20.7%)

Note: HCWs, healthcare workers; Outpatient HC, outpatient healthcare clinics; * *p* < 0.05 (Student’s *t*-test); multiple-choice options.

**Table 4 ijerph-18-12695-t004:** Attitudes towards introduction of compulsory vaccination (N = 1277).

	COVID-19 Vaccination			Occupation		
Compulsory Vaccination	Vaccinated N = 1076	Unvaccinated N = 201	*p* Value	Physicians N = 582	Non-Physician HCWs N = 542	*p* Value
Only HCWs	658 (61.2%)	5 (2.5%)	<0.001 *	378 (64.9%)	213 (39.3%)	<0.001 *
Selected population	644 (59.9%)	4 (1.9%)	<0.001 *	377 (64.7%)	204 (37.6%)	<0.001 *
Whole population	432 (40.2%)	1 (0.5%)	<0.001 *	241 (41.4%)	148 (27.3%)	<0.001 *

Note: HCWs, healthcare workers; * *p* < 0.05 (chi-square test).

## Data Availability

All data are fully available without any restriction upon reasonable request.

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
