# Peer review of "Acceptance of COVID-19 Vaccination among Healthcare and Non-Healthcare Workers of Hospitals and Outpatient Clinics in the Northern Region of Slovakia"

_ijerph, 2021, doi:10.3390/ijerph182312695_

Round 1

Reviewer 1 Report

Page 3 - Include questionnaire as an appendix rather than trying to explain in a paragraph.

Table 1 should be all on one page. It is hard to read as divided on 2 pages

Page 4, line 125: Explain tis sentence - how did you get the numbers - is it from table 2??

Page 7, line 183: Reference number is incorrect

Page 7, line 191: New statistical information introduced in discussion. This should be mentioned in results section and explained in discussion

Page 7, line 196: Reference incorrect - German study is # 20. Either fix in-text reference numbers or numbers in reference section

Page 8, line 206: reference # incorrect

Page 10: Reference numbers need to be fixed - see attachment. Ref #25 not used in text.

Author Response

Dear reviewer, 

thank you for your comments, which were very inspiring for us. We have edited the text of the manuscript according to your recommendations.

Point 1: Page 3 - Include questionnaire as an appendix rather than trying to explain in a paragraph.

Response 1: We have included the questionnaire as a supplementary material.

Point 2: Table 1 should be all on one page. It is hard to read as divided on 2 pages

Response 2: The table 2 is placed on one/same page.

Point 3: Page 4, line 125: Explain tis sentence - how did you get the numbers - is it from table 2??

Response 3: The results are presented in Table 1 (also marked in the text).

Point 4: Page 7, line 183: Reference number is incorrect

Response 4: References are correctly assigned and follow instructions.

Point 5: Page 7, line 191: New statistical information introduced in discussion. This should be mentioned in results section and explained in discussion

Response 5: We added the results to the results section and explained in discussion section.

Point 6: Page 7, line 196: Reference incorrect - German study is # 20. Either fix in-text reference numbers or numbers in reference section

Response 6: References are correctly assigned and follow instructions.

Point 7: Page 8, line 206: reference # incorrect

Response 7: References are correctly assigned and follow instructions.

Point 8: Page 10: Reference numbers need to be fixed - see attachment. Ref #25 not used in text.

Response 8: References are correctly assigned and follow instructions.

We also have edited the text of the manuscript according to your recommendations:

  • line 25 and 227: we changed "coverage" to "rates".
  • line 26: we changed "in" to "to".
  • line 40: we changed sentence structure to "Non-pharmaceutical interventions ………… national vaccination programmes are the only available tools to allow humanity to pre-pandemic life and to protect people against COVID-19".
  • line 44: we changed "vaccination" to "vaccinations".
  • line 61: we changed "toward" to "towards".
  • line 65: we changed "from 30 August to 30 September 2021" to "between 30 August and 30 September 2021".
  • line 112: we changed "accessed the study" to "were given questionnaires".
  • line 153: we changed sentence structure to "Non-physician HCWs, employees in hospitals, and employees without ………. significantly more".
  • line 164: we changed "were for introduction" to "were agreeable towards introduction".
  • line 174: we inserted "through".
  • line 180: we changed "the most fully vaccinated countries" to "the countries with highest vaccination rates".
  • line 181: we changed "are" to "were".
  • line 183: we changed "as it has only 41.83% fully vaccinated people" to "as only 41.83% are fully vaccinated people".
  • line 226: we deleted "in" before "the" to read "advocacy to the general..."
  • line 227: we changed "hesitance" to "hesitancy".

We believe that this publication will bring new insight into the issue and will take appropriate action in the professional community.

The manuscript has been 'spell checked' and 'grammar checked' (Proof-Reading-service.com Ltd, UK).

Yours sincerely

prof Viera Svihrova, MD, PhD

Reviewer 2 Report

This paper is very current. Some revisions are needed:

  • In lines 70-72, authors reported that "The study was completed by 1,277 of 4,268 employees" Can this in your opinion represent a bias in the manuscript?
  • In results section, unvaccinated people were far fewer than the other group. Can this alter statistical data?
  • In lines 184-193, it must be discussed that Covid-19 blocked several elective medical activities, causing distress to the population. Please consider this 3 important papers: Elective surgery cancellations due to the COVID-19 pandemic: global predictive modelling to inform surgical recovery plans. Br J Surg. 2020 Oct;107(11):1440-1449. doi: 10.1002/bjs.11  ----  Intracranial hemorrhage and COVID-19, but please do not forget "old diseases" and elective surgery. Brain Behav Immun. 2021 Feb;92:207-208. doi: 10.1016/j.bbi.2020.11.034.   ---   Elective cardiac surgery during the COVID-19 pandemic: Proceed or postpone? Best Pract Res Clin Anaesthesiol. 2020 Sep;34(3):643-650. doi: 10.1016/j.bpa
  • Please improve "Limitations and strengths" section.
  • In lines 224-225, authors reported that "5. Conclusions 224
    HCWs play a significant role in influencing vaccination decisions" This was not highlighted so well in the results and should be revised in the results section.
  • In lines 211-213, authors reported that "The non-physician HCWs' workplace enhances the infection risk for them, their family, and the community" Can telemedicine during covid-19 reduce this risk? discuss it. DOI: 10.1080/02688697.2020.1773399  ---  DOI: 10.31083/j.rcm.2020.04.188
  • Writings in figure 2 are very small, can they be improved?

Author Response

Dear reviewer, 

thank you for your comments, which were very inspiring for us. We have edited the text of the manuscript according to your recommendations. The manuscript has been 'spell checked' and 'grammar checked' (Proof-Reading-service.com Ltd, UK).

Point 1: In lines 70-72, authors reported that "The study was completed by 1,277 of 4,268 employees" Can this in your opinion represent a bias in the manuscript?

Response 1: The return of the questionnaire was reasonable. According to demographic data, the statistical distribution of the file was parametric.

Point 2: In results section, unvaccinated people were far fewer than the other group. Can this alter statistical data?

Response 2: The research focused on comparing the attitudes of the vaccinated and the unvaccinated employees. This is a cross-sectional study (not case-control study). For this reason, we consider that the distribution of respondents in the context does not alter the results.

Point 3: In lines 184-193, it must be discussed that Covid-19 blocked several elective medical activities, causing distress to the population. Please consider this 3 important papers: Elective surgery cancellations due to the COVID-19 pandemic: global predictive modelling to inform surgical recovery plans. Br J Surg. 2020 Oct;107(11):1440-1449. doi: 10.1002/bjs.11  ----  Intracranial hemorrhage and COVID-19, but please do not forget "old diseases" and elective surgery. Brain Behav Immun. 2021 Feb;92:207-208. doi: 10.1016/j.bbi.2020.11.034.   ---   Elective cardiac surgery during the COVID-19 pandemic: Proceed or postpone? Best Pract Res Clin Anaesthesiol. 2020 Sep;34(3):643-650. doi: 10.1016/j.bpa

Response 3: We added to discussion: The coronavirus pandemic concurrently brings issue of postponing or cancelling elective surgeries. Ultimately, this has a significant impact on patients, causing distress to the population, and negative consequences for all healthcare systems. Furthermore, postponement of elective surgery should be a carefully considered step after weighing the dangers of transmitting the infection to the patient compared to morbidity and mortality incurred by postponing or cancelling surgery. Study of COVIDSurg Collaborative estimated the total number of elective surgeries that would be cancelled worldwide during the 12 weeks. The best estimate was that 28,404,603 surgeries would be cancelled or postponed globally. In Slovakia, 4,644 surgeries would be cancelled per week during the COVID-19 pandemic [16, 17, 18].

Point 4: Please improve "Limitations and strengths" section.

Response 4: We improved „Limitations and strengths´ section.

Point 5: In lines 224-225, authors reported that "5. Conclusions 224 HCWs play a significant role in influencing vaccination decisions" This was not highlighted so well in the results and should be revised in the results section.

Response 5: We changed the HCW to physicians in the “Conclusions”. We added text to the results.

Point 6: In lines 211-213, authors reported that "The non-physician HCWs' workplace enhances the infection risk for them, their family, and the community" Can telemedicine during covid-19 reduce this risk? discuss it. DOI: 10.1080/02688697.2020.1773399  ---  DOI: 10.31083/j.rcm.2020.04.188

Response 6: We added to discussion: Hospitals and outpatient clinics increase the risk of infection both for HCWs and patients, and also for their families. Telemedicine was not used in Slovakia before the pandemic. The patient's examination took place either in a medical facility or at the patient's home. Due to these facts and the fact that non-essential medical activities are often limited during pandemic, the use of telemedicine has become an important way of managing patients' healthcare. Telemedicine has been used abroad in the past, but during the pandemic, there are more cases where specialists from various medical disciplines have started using telemedicine to care for patients. One of the several benefits of telemedicine is the reduction of the risk of COVID-19 infection through the social distancing [28, 29].

Point 7: Writings in figure 2 are very small, can they be improved?

Response 7: Figure 2 is edited.

We believe that this publication will bring new insight into the issue and will take appropriate action in the professional community.

Yours sincerely

prof Viera Svihrova, MD, PhD

Reviewer 3 Report

Ulbrichtova et al have provided their manuspript entitled Acceptance of COVID-19 vaccination among healthcare and non-healthcare workers of hospitals and outpatient clinics in the northern region of Slovakia. The paper is well written, but some changes should be added:

  • please provide the original complete questionnaire
  • please discuss in detail, how the level of predominant women up to almost 80 % influences the results of the study
  • authors should discuss vaccination availability in slovakia- are there enough vaccinations for all people? What strategies have been applied to get vaccination more "popular" so far? What might be country specific problems/reasons concerning low vaccination rates?
  • how do the authors explain this quite low rate of 41 % fully vaccinated people in slovakia?
  •  I"nterestingly, results of a German study showed 196
    92% acceptance of COVID-19 vaccination among HCWs [19]"
    - the study mentioned was performed in China- what is the correlation to Germany? please cite correctly
  • please check citation 20- in the text, citation is also not given correctly

Author Response

Dear reviewer, 

thank you for your comments, which were very inspiring for us. We have edited the text of the manuscript according to your recommendations. The manuscript has been 'spell checked' and 'grammar checked' (Proof-Reading-service.com Ltd, UK).

Point 1: Please provide the original complete questionnaire

Response 1: We have included the questionnaire as a supplementary material.

Point 2: Please discuss in detail, how the level of predominant women up to almost 80 % influences the results of the study

Response 2: We added to discussion: In general, we know that women are more represented in health services than men. In Slovakia, at the end of 2019, there were110,778 persons working in the health services, of which 23,913 (21,6%) were men and 86,865 (78,4%) were women [26].

Point 3 and 4: Authors should discuss vaccination availability in Slovakia- are there enough vaccinations for all people? What strategies have been applied to get vaccination more "popular" so far? What might be country specific problems/reasons concerning low vaccination rates? How do the authors explain this quite low rate of 41 % fully vaccinated people in slovakia?

Response 3 and 4: We added to discussion: Low vaccination rates are the result of a weak vaccination strategy. Solidarity based vaccination strategies can only be effective if people have enough reliable information and consider all risks related to vaccination as low. Nowadays, Slovakia hasn´t introduced any type of compulsory vaccination. Fortunately, with recent tightening of measures for the unvaccinated, we are seeing a slight increase in vaccination rate [4, 5].

Point 5: I"nterestingly, results of a German study showed 196 92% acceptance of COVID-19 vaccination among HCWs [19]"- the study mentioned was performed in China- what is the correlation to Germany? please cite correctly

Response 5: References are correctly assigned and follow instructions.

Point 6: Please check citation 20- in the text, citation is also not given correctly

Response 6: References are correctly assigned and follow instructions.

We believe that this publication will bring new insight into the issue and will take appropriate action in the professional community.

Yours sincerely

prof Viera Svihrova, MD, PhD

Reviewer 4 Report

Decision: major revision.

Comments:
This is an important study. It is crucial for us to understand healthcare workers’ intention to get vaccinated against COVID-19. The study examined both vaccinated HCWs and nonvaccinated ones and then investigated the motives for vaccinated and vaccine hesitancy for unvaccinated. I think it might be good to compare the two groups with common measures including motives and contributing factors to examine whether vaccination itself has dramatically changed HWCs’ attitude, risk perception, and knowledge. 

Author Response

Dear reviewer, 

thank you for your comments, which were very inspiring for us. We have edited the text of the manuscript according to your recommendations. The manuscript has been 'spell checked' and 'grammar checked' (Proof-Reading-service.com Ltd, UK).

Point: This is an important study. It is crucial for us to understand
healthcare workers’ intention to get vaccinated against COVID-19. The
study examined both vaccinated HCWs and nonvaccinated ones and then
investigated the motives for vaccinated and vaccine hesitancy for
unvaccinated. I think it might be good to compare the two groups with
common measures including motives and contributing factors to examine
whether vaccination itself has dramatically changed HWCs’ attitude, risk
perception, and knowledge.

Response: Thank you for your very inspiring comments, which may be the aim of our further research. We did not use psychometric analysis. This is a cross-sectional study that described:

  • motives to get vaccinated against COVID-19 among vaccinated HCWs,
  • reasons for hesitancy to get vaccinated against COVID-19 among HCWs,
  • attitudes towards introduction of compulsory vaccination.

We added to “Materials and methods”: Motives for vaccination were collected only among vaccinated respondents. Reasons for hesitancy were collected only among unvaccinated respondents.

We have included the questionnaire as a supplementary material.

We believe that this publication will bring new insight into the issue and will take appropriate action in the professional community.

Yours sincerely

prof Viera Svihrova, MD, PhD

Round 2

Reviewer 2 Report

A good revision was made.